# Alkyl Glycerol Ethers as Adaptogens

**DOI:** 10.3390/md21010004

**Published:** 2022-12-21

**Authors:** Ruslan Sultanov, Ekaterina Ermolenko, Tatiana Poleshchuk, Sergey Kasyanov

**Affiliations:** 1A.V. Zhirmunsky National Scientific Center of Marine Biology (Far Eastern Branch), Russian Academy of Sciences, 17 Palchevskogo Str., 690041 Vladivostok, Russia; 2Department of Normal and Pathological Physiology, Pacific State Medical University, 2 Ostryakova Ave., 690002 Vladivostok, Russia

**Keywords:** Alkyl glycerol ethers, n-3 PUFA, plasmalogens, immunostimulation, stress

## Abstract

Τhis mini-review summarizes the hematopoietic and immunostimulating properties of alkyl glycerol ethers (AGs) reported earlier in the literature available to us. The role of AGs in the nervous system and aging of the body are also briefly described. We made an attempt to consider the data in terms of adaptation. The hematopoietic, immunostimulating and antioxidant properties of AGs in a variety of experimental situations, including stress, as well as the protective action of AGs against some adaptation diseases, allow us to consider them as substances that prevent some negative effects of stress and promote adaptation. The new approach to AGs as adaptogens seems promising and opens good opportunities for their new application.

## 1. Introduction

1-*O*-alkylglycerols ethers are formed by fatty alcohols and glycerol. On average, the alkyl radical usually consists of 12–24 carbon atoms (mostly 14–18) and 1–2 unsaturated bonds, or sometimes more [1,2]. The common names of the alkyl glycerol ethers (AGs)—chimyl [16:0 alkyl], batyl [18:0 alkyl] (Figure 1), and selachyl (octadecyl) [18:1 alk-9-enyl] alcohols—are based on the names of fish species from which they were originally isolated [1,3]. The AGs structure is present in various classes of lipids: as neutral lipids in AGs and polar lipids in plasmalogen forms. Various AGs differ in the structure of the alkyl radical [1,2]. Significant amounts of lipids with a simple ether link were found in marine hydrobionts—cartilaginous fish species (sharks, skates, and chimeras) [4,5], in mollusks [6], sea stars [7], zooplankton [8] and other marine organisms, which can be useful for creation of medical preparations with a wide range of compensatory action.

## 2. About Our Experiments

This mini-review includes the results of the research conducted by the team of the Laboratory of Pharmacology of the A.V. Zhirmunsky Institute of Marine Biology (now—NSCMB) on the biological activity of AGs and their combination with polyunsaturated fatty acids in the context of the the available literature data. The last two experiments of our team were published as research articles [9,10]. In our experiments we used substances obtained from the digestive gland of gonatid squid (*Berryteuthis magister*) containing 90% AGs (the supplement “NanoMind”, Russia). An attempt was made to consider the role of AGs in terms of adaptation.

## 3. Mechanisms of AGs Action

The mechanisms of action of AGs are not fully understood. AGs bind and inhibit purified protein kinase C (PKC) in vitro [11,12]. The AGs mix from shark liver oil with various carbon chain lengths induces calcium influx in cultured human lymphocytes [13]. AGs regulate adipogenesis presumably through the γ peroxisome proliferator-activated receptor (γ-PPAR) [12]. In addition, AGs are precursors of such biologically active substances as plasmalogens and the platelet-activating factor (PAF) [12,14]. The classical scheme for the synthesis of plasmalogens in mammals (see Appendix A).

Plasmalogens, in turn, are components of cell membranes and affect their properties [12], e.g., the structure of lipid rafts, membrane fusion, and the conduction of neurons and the release of their neurotransmitters [12,15,16,17]. Plasmalogens also affect the signaling pathways associated with the membrane, protein kinase B (PKB) and mitogen-activated protein kinase (AKT/PCB). Plasmalogens act through the peroxisome proliferator-activated receptor (α-PPAR). Plasmalogens affect the homeostasis of cholesterol, ω-3, and ω-6 fatty acids (PUFA), particularly arachidonic (AA) and docosahexaenoic acids (DHA), and are also precursors of PAF, lysoplasmalogens, and 2-halo fatty aldehydes. Thus, the action of plasmalogens can be realized through the listed biologically active substances and their characteristic signaling pathways [12].

A diagram illustrating the main functions of plasmalogens in the immune and central nervous systems is given in Figure 2. Our scheme shows an attempt of graphical presentation of the mechanisms of action and properties of AGs and related substances based on the review by Dorninger et al. [12]. The study also includes data from the review by Paul, Lancaster, and Meikle [18] on the effect of plasmalogens on cholesterol metabolism, and data obtained from Kulikov, Muzy [19], Snyder et al. [3], and Facciotti F. et al. [20]. Moreover, we include the aggregate data obtained in our laboratory [21] and data from the study by Ali et al. [22]. An emphasis is placed on the effects of AGs and their related substances on the immune and nervous systems.

## 4. Oxidative Stress

It is well known that plasmalogens protect membrane lipids from oxidation [23]. The mechanism of the antioxidant action of plasmalogens is associated with the simple vinyl ether bond that has a relatively low dissociation energy and is preferentially oxidized by various free radicals and reactive oxygen species [16]. Upon interaction with negatively charged phospholipids, especially cardiolipin, cytochrome C undergoes a conformational alteration with subsequent displacement. Cardiolipin-activated cytochrome C (or plasmalogenase) catalyzes the oxidatively enabled hydrolytic cleavage of the vinyl ether linkage of plasmenylcholine and plasmenylethanolamine in the presence of hydrogen peroxide (H_2_O_2_) [24]. AGs, as a precursor of plasmalogens, act as antioxidants and participate in the binding of the reactive oxygen species [16,25]. AGs improve the oxidation-reduction status in the rat model of experimental dyslipidemia by increasing the level of catalase and normalization of the total antioxidant activity of blood plasma [26]. AGs (157 mg/kg) also restore the oxidative status of laboratory animals (rats) under acute immobilization stress by preventing catalase activity inhibition [9]. Dean and Lodhi [27] suggested that the antioxidant capacity of plasmalogens may be contextually dependent.

## 5. Aging

It has been found that long-lived animals (birds and mammals, including humans) have a lower degree of total tissue and mitochondrial fatty acid unsaturation and peroxidizability index than those of short-lived animals [28]. Pradas et al. [29] demonstrated a specific ether lipid profile that is more resistant to lipid peroxidation in centenarians. As the synthesis of alkylglycerols in peroxisomes decreases with age, the consumption of these substances is essential [30,31,32]. Age-related peroxisome degradation leads to a decrease in the activity of enzymes involved in the biosynthesis of ether lipids and β-oxidation of fatty acids [33].

Peroxisome dysfunction causes a decrease in the level of plasmalogens and docosahexaenoic acid (DHA) in aged patients with signs of Alzheimer disease [34,35,36].

AGs improve the blood lipid profile in a rat model of experimental dyslipidemia [26], as well as in old rats [10]. Ethanolamine plasmalogens have been shown to efficiently alleviate atherosclerosis via lowering cholesterol levels by suppressing farnesoid X receptor expression [37]. Atherosclerotic plaque in the aorta was reduced in treated with batyl alcohol ApoE- and ApoE/GPx1-deficient mice [38].

In this way, AGs may have a potential use in the treatment of some age-related diseases.

## 6. Nerve System

Plasmalogens are regulators of the normal activity of the nervous tissue and the brain as a whole, preventing the development of dementias, including Alzheimer’s disease and Parkinson’s disease [12,18,27,31]. An increase in β-amyloid (Aβ) levels associated with Alzheimer’s disease leads to oxidative stress in the brain and loss of peroxisomal function, while it also decreases the activity of alkylglycerone phosphate synthase (AGPS is the rate-limiting enzyme of ether lipid synthesis) and ultimately reduces the plasmalogen levels [27]. AGs prevented M1 microglial activation (including increased IL-10 and decreased IL-1β expression), contributing to the maintenance of normal neurogenesis levels within the hippocampus and normalizing working memory in mice neuropathic pain model [21]. Scallop-derived plasmalogens inhibit the lipopolysaccharides mediated endocytosis of toll-like receptors (TLR4) and the downstream caspases activation. The TLR4 endocytosis and the caspases activation strictly control the pro-inflammatory cytokine (IL-1β and TNF-α) expression. Therefore, plasmalogens attenuate the microglial activation by maintaining the endocytosis of TLR4 [22]. Moreover, AGs can display their activity through an additional pathway via PAF.

## 7. Hematopoiesis, Immunity and Inflammation

The use of AGs enhances the body’s protective functions—hematopoietic and immunostimulating activities [1,2,11,18,39].

As precursors of PAF, AGs have an indirect effect on blood cells [1]. PAF is a powerful bioregulator, and the profile of its biological activity is not limited only to the effect on platelet aggregation [3,12,40]. PAF is involved in the production of cellular phlogogens, and thus becomes involved in the formation of the body’s immune response [19].

The level of erythrocytes in sows increased after a 5-week intake of shark liver oil [41]. Osmond et al. found three changes of erythropoiesis in guinea pigs, induced by moderate doses of batyl alcohol after five days of subcutaneous administration. This application resulted in (1) an increase in the blood reticulocytes, (2) an increase in the marrow reticulocytes and (3) a trend to increase in the absolute count of nucleated erythroid cells of the marrow [42]. Iannitti and Palmieri [1] summarized that optically active and racemic batyl alcohol stimulates erythropoiesis, thrombopoesis and granulopoesis, and chymil alcohol stimulates haemopoesis, while selachyl alcohol had no haemopoetic activity. In our laboratory research, AGs administration resulted in an increase in the erythrocyte level in rats [26] and restored the red blood cell mass in aging rats [10]. Platelets increased in the dislipidemia model [26] and decreased in aging rats (compared with aged control) [10]. Blood clotting time increased with the application of AGs [26]. In addition, AGs partially inhibited [3H]-serotonin release induced by PAF, while they did not modify spontaneous or thrombin-induced release [43]. A combined effect of AGs and n-3 PUFAs on erythropoiesis and thrombopoiesis is more pronounced, their action is most likely induced through common pathways implemented by plasmalogens and/or PAF. Only a combined administration of n-3 PUFAs and AGs contributed to a tendency of an increase in the average hemoglobin content in one red blood cell. The combined introduction of n-3 PUFAs and AGs contributed to a maximum decrease in the platelet count [10].

Administration of AGs increased the level of leucocytes and lymphocytes in aged rats [10] and lymphocytes in a rat chronic stress model (200 mg/kg during 6 weeks) (unpublished data). White blood cells count, IgG and lymphocytes significantly increased, while neutrophils significantly decreased in aging patients after surgical and AGs treatment (a dose of 500 mg twice a day over 4 weeks) [44]. The greatest increase in the number of leukocytes was obtained at a separate use of AGs or AGs in combination with n-3 PUFAs. In contrast, lymphocytes showed a stronger response to an AGs-enriched diet [10]. Therefore, AGs must be the substances with direct and indirect pathways that participate in the processes associated with immunity. According to literature data, AGs can modulate immune responses by boosting the proliferation and maturation of murine lymphocytes in vitro [45]. Studies on the immunostimulatory action of alkylglycerols suggest a primary action on macrophages. The process of macrophage activation was demonstrated in both synthetic and natural AGs [11]. AGs stimulate lysosomal activity and the synthesis of nitric oxide, increase the level of reactive oxygen species and interleukin-6 expression and stimulate in the RAW264.7 murine macrophage cell line at concentrations of 0.1 to 5 µg/mL [46], thereby demonstrating the immunostimulating activity of AGs in vitro. Shark liver oil is able to increase lymphocyte proliferation in rats [47] and the levels of leucocytes, lymphocytes, neutrophils, monocytes and IgGs in the blood of piglets from supplemented sows [48]. In their report, in a closed model study, Benzoni and coauthors [48] indicated that the administration of fatty acids and AGs composition to sows in late gestation and lactation improved passive immunity transfer to piglets. They suggested that blood complement activity seems to be an accurate indicator of immuno-stimulating additive efficiency. Peripheral blood granulocytes, IgG and IgM elevated in pups from the rat dams fed AGs [49]. Mice deficient in the peroxisomal enzyme glyceronephosphate O-acyltransferase, essential for the synthesis of ether lipids, indicated a significant alteration in maturation in the thymus of stimulated semi-invariant natural killer T cells and that there were fewer of these cells in both the thymus and peripheral organs. Ether-bonded mono-alkyl glycerophosphates, the precursors and degradation products of plasmalogens and their synthetic analogs stimulated semi-invariant natural killer T cells [20]. Shark liver oil increased specific vaccination-associated antibodies in the blood of sows [41]. In the bone marrow of guinea pigs, the population of small lymphocytes and transitional cells was increased by batyl alcohol, while granulocyte precursors at the metamyelocyte stage were decreased by selachyl alcohol [42]. Therefore, AGs can be involved in both leukopoiesis and in the processes of the maturation and activation of lymphocytes and macrophages.

Moreover, since the hematopoietic and immunostimulatory effects of AGs are observed in different mammals, it is reasonable to assume that these effects are similar across the entire class Mammalia.

Pathophysiological conditions involved chronic inflammatory processes are linked with decreased levels of plasmalogens [50]. Imbalances of major lipid signaling pathways contribute to disease progression in chronic inflammation [23]. Platelet-activating factor (PAF) is versatile inflammatory mediator. 2-chloro fatty aldehydes modulate inflammatory and immune processes. Plasmologens are also involved in the metabolism of PUFAs, which include both pro-inflammatory (AA and its products) and anti-inflammatory (DHA). [12].

## 8. Other Properties of AGs

Under the action of AGs and the combined use of AGs with n-3 PUFAs, the concentration of 16:0 dimethyl acetals (DMA) in rat liver increased, as evidenced by the inclusion of chymyl alcohol plasmalogens (C16:0) into the biosynthesis. Under the action of n-3 PUFAs and the combined use of n-3 PUFAs with AGs, the concentration of docosahexaenoic acid (DHA) in rat liver increased [10]. The introduction of AGs resulted in a reduction of triglycerides in the blood serum of rats [26].

In the gastroduodenal tract, AGs reduce stress-related ulcers in rats [9]. Also, data from the literature indicate that shark liver oil supplementation can improve the amelioration of acetic acid-induced ulcerative colitis in rats due to its antioxidant effects [51].

## 9. Conclusions

It can be assumed that different mechanisms of action of AGs are triggered by different doses. The protective action of AGs on the gastric mucosa is observed at a low dose (15 mg/kg). The middle doses (157–200 mg/kg) of AGs increase catalase activity under acute stress [9] and lead to a rise of lymphocytes in rats [10]. A high dose (400 mg/kg) causes an increase in the level of erythrocytes [26].

Selye [52] considered anemias, gastrointestinal ulcers, and ulcerative colitis as diseases of adaptation. One of the classic signs of stress described by Selye is the involution of the lymphoid organs. The author also lists systemic infections resulting from the reduced resistance that occurs in such cases, and aging among the diseases of adaptation. Since all these and some other states are corrected in one or another way by AGs, we can assume their participation in the mechanisms of adaptation. In addition, any alteration in homeostasis leads to an increased production of free radicals, while chronic stress potentiates oxidative stress [53]. AGs, as mentioned above, contribute to the normalization of the redox status, also under acute stress. Thus, the study of AGs in the light of the concept of adaptation, one of the fundamental theories of modern biology and medicine, is interesting and worth further studying.

The literature and our own research data have shown that the hematopoietic, immunostimulating and antioxidant properties of AGs described in the literature occur in various experimental situations, including different experimental stress models.

We can thus draw the following main conclusions:AGs prevent many of the negative effects of aging.Plasmalogens are regulators of the normal activity of the nervous tissue and the brain as a whole and prevent the development of dementias.AGs stimulate hematopoiesis, which contributes to the body adaptation to various conditions, including extreme ones.AGs improve the body immune status.A novel approach to AGs as adaptogens seems promising and creates multiple opportunities for their potential application.

## 10. Outlooks

AGs reduce many of the negative effects of stress. Therefore, we consider it promising to study AGs in order to develop practical recommendations for their use in stressful situations, adaptation to extreme environmental conditions, and also in the treatment of gastrointestinal ulcers. A promising direction for further research on AGs and plasmalogens is a deeper study of their role in the development of the nervous system and neurodegenerative diseases, including Alzheimer’s disease. Therefore, a deep study of the role of AGs and plasmalogens can form the basis for solving important social problems, which include effective adaptation to various conditions, the fight against neurodegenerative diseases and the adverse changes that accompany aging.

## Figures and Tables

**Figure 1 marinedrugs-21-00004-f001:**
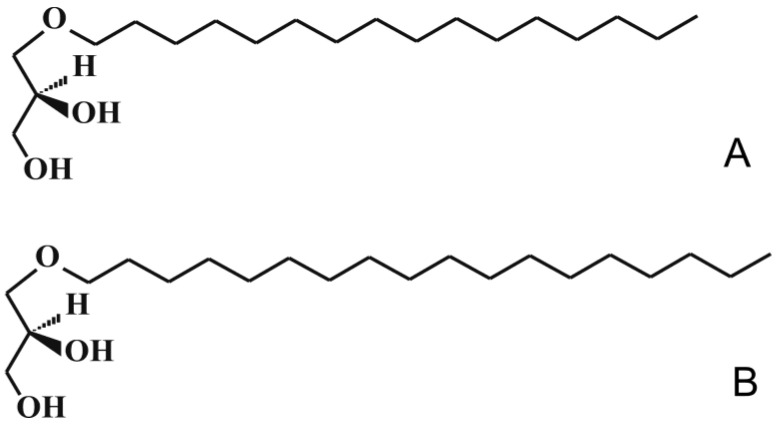
The chemical structure of alkylglycerols. (**A**)—chimyl alcohol, (**B**)—batyl alcohol.

**Figure 2 marinedrugs-21-00004-f002:**
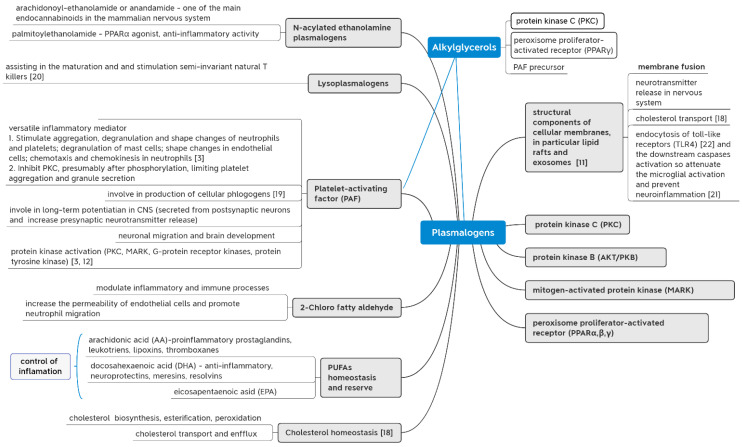
The mechanism of action of plasmalogens. The scheme is based on the data from the review by Dorninger et al. [12]. It includes data from the review by Paul, Lancaster, and Meikle [18] on the effect of plasmalogens on cholesterol metabolism, and data on PAF reviewed by Snyder, Lee, and Wykle [3] about PAF. We also used the study by Facciotti et al. [20], Kulikov and Muzy [19], and data from Tyrtyshnaia et al. [21] and Ali et al. [22]. An emphasis is placed on the effect of AGs on the immune and nervous systems.

## Data Availability

Not applicable.

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
