# Peer review of "Alkyl Glycerol Ethers as Adaptogens"

_marinedrugs, 2022, doi:10.3390/md21010004_

Round 1
Reviewer 1 Report
This is an excellent manuscrtipt. I have only 2 suggestions.
First, there is extensive discussion of plasmalogens in the manuscripot but many readers do not fully understand the uniwue structural attribites of the membrane lipids. In think an addition to Fig. 1 showing the conversion of the AGs to plasmalogens would be very useful.
Seond, Section 4. should be AGs in the nervous system
Author Response
Dear Reviewers,
in the attached files answers to your questions and comments We took them into account to improve our manuscript.

Reviewer 2 Report
Alkyl glycerols are not well known molecules whereas they have several biological activities. This review is therefore of interest. However, major points are missing and major modifications are required.
Minor:
Figure 2 is of poor quality. A new figure must be provided.
Major
* Nothing is said on the biogenesis of alkyl glycerols. This point must be presented. If possible a figure must be added.
* Nothing is said on the impact of acid glycerols on oxidative stress, and inflammation. This is a important point which must be considered.
* Acyl glycerols may have some interest in the treatment of some age related diseases, especially atheroscleosis. This paragraph must be developped.
Overall, the manuscript must be improved as suggested and new references must be added.
Author Response
Dear Reviewers,
In the attached filesanswers to your guestions and comments. We took them into account to improve our manuscript.

Round 2
Reviewer 2 Report
The present review provides lot of information on alkyl glycerol ethers (AGs) and has been improved .
However, minor revisions are still required (mandatory) and the paper must be modified and resubmitted.
1) Figure 2 is of very poor quality and must be improved.
A new one is required. On the present figure it is very difficult to read the text.
2) Paragraph 5, Aging, line 183-184
The following ref must be added
Lizard G, Rouaud O, Demarquoy J, Cherkaoui-Malki M, Iuliano L. Potential roles of peroxisomes in Alzheimer's disease and in dementia of the Alzheimer's type. J Alzheimers Dis. 2012;29(2):241-54. doi: 10.3233/JAD-2011-111163. PMID: 22433776.
Author Response
Please see the attachment. We have attached the answer below.
